# Value of Three-Dimensional Imaging Systems for Image-Guided Carbon Ion Radiotherapy

**DOI:** 10.3390/cancers11030297

**Published:** 2019-03-02

**Authors:** Yang Li, Yoshiki Kubota, Mutsumi Tashiro, Tatsuya Ohno

**Affiliations:** 1Graduate School of Medicine, Gunma University, Showa 3-39-22, Maebashi, Gunma 371-8511, Japan; m1820051@gunma-u.ac.jp; 2Gunma University Heavy Ion Medical Center, Gunma University, Showa 3-39-22, Maebashi, Gunma 371-8511, Japan; y_kubota@gunma-u.ac.jp (Y.K.); tashiro@gunma-u.ac.jp (M.T.)

**Keywords:** carbon ion radiotherapy, image-guided radiotherapy, in-room CT, anatomical changes, hypofractionated radiotherapy, adaptive radiotherapy

## Abstract

Carbon ion radiotherapy (C-ion RT) allows excellent dose distribution because of the Bragg Peak. Compared with conventional radiotherapy, it delivers a higher dose with a smaller field. However, the dose distribution is sensitive to anatomical changes. Imaging technologies are necessary to reduce uncertainties during treatment, especially for hypofractionated and adaptive radiotherapy (ART). In-room computed tomography (CT) techniques, such as cone-beam CT (CBCT) and CT-on-rails are routinely used in photon centers and play a key role in improving treatment accuracy. For C-ion RT, there is an increasing demand for a three-dimensional (3D) image-guided system because of the limitations of the present two-dimensional (2D) imaging verification technology. This review discusses the current imaging system used in carbon ion centers and the potential benefits of a volumetric image-guided system.

## 1. Introduction

The integration of imaging technology with a linear accelerator has led to the introduction of image-guided radiotherapy (IGRT) and the popularity of this technology has grown over recent years. A survey reported by the American Society of Radiologic Technologists showed that more than one-third of participants use cone-beam computed tomography (CBCT) technology [1], and over 80% of radiation therapy institutions have used IGRT systems in the United States since as early as 2010 [2]. Recently, image-guided particle radiotherapy (RT) has attracted increasing attention. In particular, carbon ion radiotherapy (C-ion RT), because of its excellent dose distribution and high biological effects [3], has enormous clinical advantages over conventional RT and even proton RT [4,5]. More than 20,000 patients were treated by C-ion RT by 2016, and nine centers were built given its ideal clinical outcomes according to the Particle Therapy Co-Operative Group (Figure 1). Interestingly, those results were based on a patient positioning system with two-dimensional (2D) imaging verification technology. Although this system facilitates an excellent patient setup with minimal translational and rotational residual errors (<0.5 mm and 0.2°, respectively) [6], better results can be expected when a three-dimensional (3D) image-guided system is used, such as computed tomography (CT). Volume imaging modalities can help detect geometrical changes including tumor and surrounding healthy tissues. Thus, inter- and intra-fractional errors can be corrected, which is particularly important for maintaining the robustness of dose distribution for C-ion RT. Because the dose coverage of C-ion RT is very sensitive to those changes, small changes in anatomical structure may cause large deviations in dose distribution. Thus, this effect should be carefully monitored. To confer potential benefits from a CT image-guided system on treatment via C-ion RT, numerous simulation studies have obtained consistent results—using CT images can obtain a better dose distribution with less positioning errors than conventional orthogonal X-ray images [7,8,9,10]. Therefore, volumetric image-guided technology should be paid particular attention. This review aimed to describe the value of a CT imaging system in image-guided C-ion RT.

## 2. Present Verification System in Carbon Ion Radiotherapy

Using orthogonal X-ray images for patient positioning verification is the standard procedure for C-ion RT in most centers (Figure 2). In usual clinical practice, positioning error is measured by comparing some positions on a digital radiography (DR) image with the corresponding positions on a digitally reconstructed radiography (DRR) image from the treatment planning CT. Bone structural matching has been the most-employed matching technique in particle therapy [11]. Because matching with dense structures such as bone structures is important for determining the penetration of C-ion beams, bone matching has always been used as the best method for realizing a better dose distribution. Compared with 3D imaging techniques, less time is required for planar kilovoltage images. Moreover, this approach offers a lower dose to the patient. We can also achieve real-time tumor or fiducial marker tracking in extracranial sites by the fluoroscopic mode. However, the limitations of 2D imaging systems are as follows: (1) they are unable to provide sufficient anatomical information such as distinguishing tumors and soft tissues, (2) they are unable to be directly compared with planning CT images, and (3) they are unable to calculate dose distribution for re-planning.

## 3. 3D Imaging System

Modern irradiation procedures have entered the 3D era—3D/4D patient setups, 3D delineation of the target region, 3D/4D treatment plans, and 3D verification. Volumetric images can provide more information of anatomical structures, not only images of body outline or bone anatomy, but also soft tissues, with a good estimate of internal tumor position. The specific process is commonly used as follows: A CT scan is performed on patients before treatment and then the acquired CT images are registered with the original planning CT images to correct positional errors. Because of a direct comparison between CT images, deviations derived from mismatches between DRR and DR images can be minimized. However, this would require a few more minutes for the acquisition, reconstruction, and registration than needed in the 2D verification process. The advantages of the 3D imaging system are explained below.

## 4. Precise Positioning

Positioning deviation is one of the main sources of errors. The reproducibility of positioning is crucial for the application of ion beams to ensure calculated dose deposition. Using skin markers or bone matching, 2D imaging verification systems usually have high setup accuracy. For head and neck sites, no significant difference was observed between 2D and 3D imaging techniques [12,13]. However, for tumor sites in the thorax and abdomen, planar images offer limited information. For example, using CT imaging technology, setup errors could be reduced by 28% for patients with prostate cancer and 20–50% for patients with breast cancer compared to those with 2D X-ray images [14,15]. Similar results were observed in other sites such as lung cancer and esophageal carcinoma [16,17]. Those errors are primarily caused by the poor contrast in soft tissue and relative motion between the tumor and the reference organ (usually bone). In most cases, they cannot be clearly displayed on DR images. In addition, inherent limitations of matching deviations (such as translation and rotation errors) between DR and DRR images may cause uncertainties [18].

The introduction of fiducial markers addresses the problem that DR is largely restricted to bone structures. The markers are usually implanted in the observed organ or in the tumor tissue itself. Thus, those clearly displayed on the DR image can help determine the location of the tumor. This method can improve the accuracy of target coverage compared with bone matching [9]. However, uncertainties still exist because of the relative displacement between markers and tumors [7]. The average fiducial marker displacement was larger than 10 mm in approximately 33% of lung cancers [19]. Moreover, the marker itself influences the dose distribution in particle therapy [20,21,22]. In addition, it is an invasive technology. Therefore, soft tissue image-guided technology based on CT seems to be a more reliable and less invasive method to monitor deviations during treatment.

## 5. Precise Radiation Delivery

Deviations of dose distribution in the target region may still occur even with a good patient setup by bone matching. Table 1 shows the dose distributions with different matching methods [9,10,23]. Compared with tumor and marker matching, bone structural matching may decrease dose distribution. This is mainly caused by relative tumor displacement and/or changes in the water-equivalent path length (WEL) along the beam path. Unlike photon therapy, WEL is an important factor which determines the dose distribution. Factors that lead to WEL changes can affect the dose distribution. Even in a short-course therapy, these changes should be taken into consideration. Our center reviewed tumor displacement using simulator CT for 50 patients with stage I lung cancer treated with four-fraction C-ion RT from 2016 to 2017 (Figure 3A) [8,10]. Large displacements were observed in all directions, especially in anterior–posterior (AP) and superior–inferior (SI) directions. However, the dose distribution of gross tumor volume was significantly decreased when tumor displacement was larger than 7.5 mm, as reported by Irie et al. [8]. Importantly, this value may become smaller when considering the dose coverage for the clinical target volume (CTV). Other important factors that must be considered are the changes in the organs surrounding the tumor, such as the chest wall thickness, gastrointestinal gas volume, and organ deformation. These changes may affect the abovementioned WEL and lead to a decrease in the dose distribution (Figure 3B). For patients with prostate cancer, large inter-fractional deviations have been observed in rectum gas and prostate position [24], and daily monitoring for changes in organs was recommended to maintain good dose coverage and spare the normal tissues, even for the patients after prostatectomy [25]. For abdominal tumors, Houweling et al. [26] evaluated the effects of daily changes of gastrointestinal gas volume on cumulative dose in nine patients with pancreatic cancer. The internal clinical target volume coverage decreased by 17% and 10% for bone and marker positioning verification, respectively. Therefore, obtaining information regarding daily anatomical changes appears necessary to assess the actual dose in the target region during C-ion treatment, even for short-course treatment (less than 2 weeks). However, this appears insufficient. The intra-fraction changes in gastrointestinal gas volume also affected the target dose distribution in patients with pancreatic cancer [27]. Therefore, particular attention must be paid to the effects of intra-fractional anatomical changes on dose distribution, which are further discussed below. In the context above, a prospective study using in-room CT to confirm the accumulated dose for mobile tumors throughout the treatment period is being conducted at Gunma University Heavy Ion Medical Center (GHMC). In summary, both inter- and intra-fractional anatomical changes can significantly affect the dose distribution in C-ion RT. This emphasizes the importance of predicting the actual accumulated dose at different stages in each fraction by daily CT images obtained before treatment.

## 6. Planning Target Volume Margin Reduction

Considering systematic and random uncertainties, we have to expand the planning target volume (PTV) margin to ensure that the target region is within the radiation field. However, this is a compromise as it may cause more exposure to healthy tissues. In photon therapy, 3D verification possibly reduces the PTV margin. Den et al. [28] studied 1013 CBCT scans to demonstrate the potential for IGRT to enable margin reduction in head and neck cancer. With daily online volumetric guidance, the PTV margin could be reduced by 50% (from 5 mm to 2–3 mm). For prostate cancer, Ariyaratne et al. [29] evaluated the dosimetric effects of frequency imaging and the PTV margin during IGRT. Compared with weekly online imaging, the PTV could be reduced from 7 mm to 5 mm with daily online CBCT. For C-ion RT, an internal margin calculated by four-dimensional computed tomography (4DCT) according to tumor motion is commonly used for a patient-specific PTV margin [8,9,10,11]. However, the PTV margin may still not suffice considering the inter-fractional anatomical changes. Sakai et al. [10] conducted a retrospective study to evaluate the effects of matching methods on dose distribution for stage I lung cancer. All images were acquired from a simulation room CT. To maintain the robustness of dose distribution for unacceptable cases (CTV V95% < 95%), the estimated required margin is 8 mm for bone matching and only 3 mm for tumor matching. Similar to photon therapy, C-ion RT with a wider PTV margin causes excessive dose delivery to normal tissues. In addition, C-ion RT requires both distal and proximal margins, which makes the treatment planning more complex. Therefore, monitoring daily geometrical changes seems critical for maintaining a tight margin for C-ion RT. In addition, considering the intra-fractional beam range variations, the internal target volume may not be useful for C-ion RT. Thus, the field-specific target volume (FTV) was introduced to compensate for intra-fractional range variation, and attained improved dose coverage of the target region for patients with lung, liver, and pancreatic cancer [27,30]. Using FTV clinically requires a high reproduction of intra-fractional motions. However, many contingencies may occur during the treatment; for example, the breathing patterns may vary. Thus, inter-fractional changes in tumor and normal tissues should also be taken into consideration. Therefore, a good estimate of these changes is necessary for adjusting the FTV. Unfortunately, a single pre-treatment 4DCT is not always representative of actual respiratory motion [31]. Daily 4DCT images are recommended to address this problem. As research continues, volumetric imaging systems must play a significant role in margin settings in C-ion RT.

## 7. Hypofractionated C-Ion Radiotherapy and Motion Management

Hypofractionated C-ion RT has been developed in phase I/II clinical studies and has achieved successful clinical results for many types of tumors in Japan [4,5,32]. Given the high dose delivered to the target with few fractions, 3D image-guided technology is necessary for precise dose localization, especially for treatments with very few fractions, such as for early stage lung cancer (from 18 fractions to a single fraction) and hepatocellular carcinoma (from 15 fractions to two fractions) [33,34]. Reducing the uncertainties caused by internal organ motions (such as respiratory, cardiac, and peristaltic motions) is very important for a successful hypofractionated C-ion RT. Currently, respiratory gated-CT and 4DCT have been widely used in C-ion RT to reduce deviations caused by respiratory motion [35]. Respiratory gated-CT, where images are collected at a certain gated window to reconstruct a CT, can obtain a relative stable and reproducible target position for treatment. Also, 4DCT, where CT images are reconstructed at each respiratory phase, allows a good estimation of the target motion in different respiratory phases. The gated treatment includes two types of tracking systems—external surrogate motion tracking and internal tumor tracking. The former commonly uses a camera to capture motions of the patient surface or artificial markers placed on the patient abdomen. The latter system is based on fiducial marker tracking by fluoroscopy. Both technologies require high correlation between the external respiratory signal or fiducial marker and the internal tumor motion, which determines the quality of the treatment. However, not all cases are satisfactory [20,36]. Deviations from mismatch or tumor displacement may still occur during treatment. To decrease the frequency of those errors, many markerless tumor-tracking techniques based on data training (machine learning) have been developed [37,38,39]. Promisingly, they have been successfully implemented by clinically combining with phase-controlled-rescanning (PCR) C-ion RT technology for mobile tumors (lung and liver cancer) with an extremely high gating accuracy [40]. The safety of these techniques was also confirmed in a follow-up study [41]. In addition, a rapid PCR technology is available, which can reduce the gated treatment time by half of that in conventional PCR (from approximately 110 s to 53 s) [42]. However, limited by the low contrast of 2D imaging, whether this system can be used for pancreatic cancer patients needs further study. Although current CT imaging capabilities cannot provide real-time 4D information, the introduction of 3D/4D in-room imaging systems may offer another way to detect the deviations such as mismatch between external signals and internal tumor movement. Therefore, combining 2D and 3D imaging modalities appears to be a good solution to maintain the robustness of respiratory-control treatment. In this way, a real-time patient-specific margin based on tumor motion may be obtained for hypofractionated C-ion RT, which may bring more promising clinical results. 

## 8. In-Room Computed Tomography

In recent years, the most widely used 3D imaging modality has been CBCT, especially in photon centers [43,44]. CBCT can be directly integrated with a linear accelerator, which enables daily monitoring of patient positioning and tumor displacement. Compared with C-ion RT, 3D verification systems develop relatively fast in proton therapy. Some proton therapy centers have adopted in-room CT technologies such as CBCT or sliding CT units [45]. Technology has also made rapid progress in C-ion RT. A superconducting rotating-gantry with irradiation angles of over ±180° is available [46]. Although it is currently impossible to integrate a CT scanner directly into the C-ion gantry, this issue is expected to be solved in the future.

Today, a CT-on-rails system is an optimal choice for C-ion centers. It does not require the redesigning of the original RT accelerator, and the space requirement for the treatment room is low. This system is used in some centers such as GHMC and the Kanagawa Cancer Center; others include the Shanghai Proton and Heavy Ion Center and the Osaka Heavy Ion Therapy Center. The CT scanner is usually installed opposite the accelerator (other angles are also allowed). The scanner can move along the guideway, called CT-on-rails. In this way, CT scanning and beam irradiation can share the same treatment couch. The treatment process is shown in Figure 4. As an important advantage, it offers diagnostic-quality images with superb soft tissue contrast. Thus, we can delineate the target volume as usual with these images. Superior-quality images can greatly improve the setup accuracy over planar imaging when alignment to soft tissue is desired [23,47,48,49]. Compared with the limited scan length of the CBCT, wide-range CT images are easy to obtain. Moreover, these images can be used to correlate with daily verification images obtained during treatment; in this way, online or offline processing can be performed [50]. It becomes easier for dose calculation to develop a new plan or evaluate the effects of anatomic structural changes on dose distribution using daily CT images. Thus, the accumulated dose for the entire treatment can be determined using the deformable image registration method [24,26]. CT imaging methods have also been proposed to measure the stopping power, which can serve as a planning dataset and online measurement of changes. Kubota et al. [51] developed a method to predict the dose distribution by replacing the stopping power ratio for inter-fractional anatomical changes. It is a simple method of obtaining more information from treatment planning CT images. However, this method might be unreliable when tumor displacement and WEL changes become large. Therefore, daily imaging verification is still an optimal way to assess the dose distribution throughout the treatment. This provides guidance for optimizing or adjusting treatment plans. However, when using CBCT, given its relatively low image quality, dose calculation for particle therapy becomes more complicated [52,53].

Concerning the short-term application of in-room CT at C-ion centers, no results on its clinical benefit have been published yet. Most studies are based on simulation research. CT images are acquired from either simulator CT or other medical centers. The latest study based on CT-on-rails in proton therapy was published by Maeda Y. et al. [49]. They analyzed 375 sets of daily CT images from 10 patients with prostate cancer. The smallest positioning deviations were found by prostate–rectum boundary matching, as compared to other conventional methods (bone matching and prostate matching) with daily CT images. In this way, the rectal dose could be reduced effectively. This suggests that diagnostic-quality CT images have great potential to bring about more accurate and individualized matching methods. However, the elucidation of the potential effects on clinical outcomes requires further study. For example, Wu et al. [54] established an organ-at-risk evaluation system based on changes in CT numbers to predict radiation damage.

There are also some limitations for in-room CT. First, because the CT scanner and beam delivery system are in different positions with a certain distance between them, the couch has to be moved between them. This may lead to new errors (from the slight movement of the patient or movement caused by driving the couch, etc.) that should be considered, even though they might be very small [24]. Obviously, this increases the treatment time, radiation dose, and operational complexity. Therefore, experienced radiation technologists are required to give adequate guidance and training to patients. However, as the technology advances, diagnostic-quality CT and RT equipment will be integrated directly in the future. Then, it would ideally possible to perform patient setup, positioning verification, even treatment on the same couch without additional movements. This would greatly simplify the treatment process and lead to fewer errors.

## 9. Image-Guided Adaptive C-Ion Radiotherapy

When using in-room CT, repositioning can align the target with its position on the planning CT image. However, this may not be enough to restore the planned dose distribution because of organ deformation such as tumor shrinkage/enlargement or structural changes in surrounding organs along the beam path. Adaptive radiotherapy (ART) in such a treatment technology is triggered by daily CT. It aims to improve the target dose coverage or avoid excessive dose to the organs at risk by adjusting the treatment plan according to daily anatomical changes. Recently, ART has been actively developed both in the photon and particle fields. Numerous studies have confirmed that it implements more accurate dose delivery for most tumors [50]. Its biggest advantage is its quick response to anatomical changes before each fractional irradiation. Thus, sufficient 3D imaging information is essential for ART implementation, especially for online ART. The ideal image dataset for re-planning needs to meet the requirements of deformable registration and dose calculation. Currently, ART is typically based on CBCT or CT-on-rails; the former is dominant because of its wide installation. However, because of the abovementioned limitations, CBCT is not considered a good imaging modality for online ART in some sites such as the head and neck, abdomen, pelvis [55,56], and even in the lungs [57]. However, a recent report argued that CBCT can help provide similar clinical indicators to those using rescan CT for lung cancer patients [58]. However, the workflow is complicated, and the uncertainties in deformable registration between planning CT and CBCT images may reduce its reliability. Therefore, high-resolution CT seems to be more suitable for clinical ART implementation. At present, most feasibility studies for adaptive C-ion RT are based on simulator CT. Compared with the conventional plan, daily re-planning could significantly improve the target dose coverage while sparing healthy tissues, especially for large tumors [59,60,61]. Although prospective studies are lacking, these simulation studies do provide a theoretical basis for further investigations.

However, many obstacles make it difficult to perform ART in clinical practice, such as the lagging technology update because of its high cost. Unlike photon facilities commonly equipped with a multi-leaf collimator, current facilities with passive scatter technology need a patient-specific collimator and compensating filters to shape the broad beam and adjust the distal conformity according to the shape of the tumor [62]. Generally, a 1-week time requirement is common before the first treatment delivery. Selecting a similar-shaped device directly from the “library” saves time, but this may decrease conformity and even dose distribution. Another strategy for ART is as follows: Two or three additional CT scans are performed to detect changes in tumors during the treatment, and a new plan is made when necessary. This method might improve the dose distribution when big changes in the tumor are observed [63]. However, it increases the total time of treatment by 2–3 weeks. Usually, prolonging the treatment time means more potential errors. Thus, it is difficult to perform ART with the burden of device manufacturing. The active beam scanning system, however, does not bring such hardware limitations. It enables the delivery of intensity-modulated dose distribution by changing the weight of each pencil beam [64]. Moreover, rapid planning based on the original plan is possible (6 min only) [65]. Nevertheless, high-resolution CT scanning and repositioning have greater time requirements (usually more than 15 min). A method which includes segment aperture morphing and segment weight optimization has been proposed, which helps to complete the intact RT process in 10 min [25]. However, whether this method works for particle RT requires further investigation. In addition, due to the narrow time window for quality assurance (QA) issues, it seems infeasible to perform a delivery QA before the treatment. Therefore, introducing a simple but reliable QA procedure is necessary for clinical practice. Finally, current evidence is very limited when it comes to offering guidance for clinical implementation; thus, more prospective studies with large samples are needed to maintain its robustness during treatment. Although much experience from conventional therapy can be considered as a reference, further study is still necessary to address the varying physical properties. Thus, online adaptive C-ion RT will need more time in clinical practice. Promisingly, the application of volume imaging systems greatly promotes the development of this technology. It will certainly be a hot spot for research in the following years.

With daily CT images a series of offline processes can be performed at present, called offline ART. Images acquired in the first few fractions are used to determine inter-fractional changes in patient positioning and confirm the factors that affect the target coverage and the sparing of normal tissues. After several fractions, patient-specific profiles can be created by defining the level of movement as well as tissue and organ changes. In this way, a more accurate positioning can be implemented, and a patient-specific PTV margin can thus be derived to minimize the dose to the surrounding normal tissues. Ghilezan et al. [66] reported that high tumor control can be achieved without increasing normal tissue toxicity for prostate cancer using the offline approach; however, compared with online ART, it has limitations regarding its response to tumor motions [61].

Magnetic resonance imaging (MRI)-guided ART is an emerging technique in the photon field. Compared with CT, it enables a better management of inter-fractional variations because of its remarkable soft tissue contrast and continuous imaging capability [67]. It also has a real-time tumor-tracking capability that addresses intra-fractional motion [68]. With fewer uncertainties, more accurate C-ion RT can theoretically be performed. This may further increase the indications of hypofractionated C-ion RT. The capability of MRI to provide functional imaging is another important advantage. It enables the use of imaging biomarkers to indicate a treatment response, which offers a completely new method to modify the treatment [69]. Considering the high biological effects of C-ion RT, MRI is potentially helpful in identifying responders compared with non-responders during treatment. In this way, real-time MRI-guided C-ion RT can be performed and can lead to better clinical outcomes.

## 10. Conclusions

Carbon ion therapy is an advanced technique that is characterized by excellent conformal dose distribution known as the Bragg Peak [70]. In comparison with photon therapy, particle therapy has numerous advantages because of its unique radiobiological properties. However, advanced image-guided technologies are necessary to guarantee these “advantages” not being affected by various uncertainties during treatment. Owing to some limiting factors, during-treatment assessments in carbon ion institutes develop more slowly than those in photon therapy centers, where a 3D imaging system is routinely used for daily alignment and positioning. Present orthogonal X-ray systems have limited the development of C-ion RT to some extent. Hypofractionated C-ion RT is being widely adopted as a more efficient treatment strategy. The fractional proposal in stage I lung cancer has changed from 18 fractions to four fractions, and even to a single fraction [33]. Adaptive C-ion RT is also a promising technology that enables the significantly improvement of the accuracy of the treatment by re-planning. These advanced modalities call for more advanced imaging technology supports. Multi-image technologies integration represents the direction of the development of IGRT in the future [71,72]. This necessitates speeding up the popularization of 3D imaging technology in C-ion RT centers. In this way, more experience can be gained to improve this technology and pave the way for integration with other imaging modalities. Promisingly, the size of the carbon ion facility has been designed to be significantly smaller than before, and scanning beams and rotating gantries have also become more available. The benefits from these technologies with volumetric imaging modalities are worth looking forward to.

## Figures and Tables

**Figure 1 cancers-11-00297-f001:**
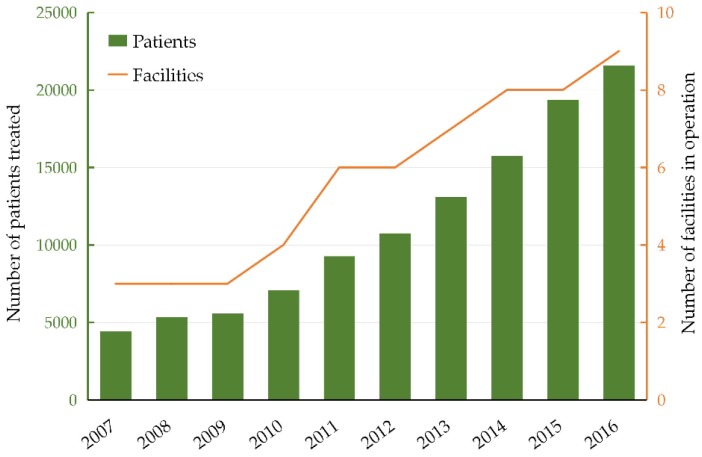
The number of patients treated by C-ion radiotherapy and the number of facilities in operation during the past 10 years (2007–2016).

**Figure 2 cancers-11-00297-f002:**
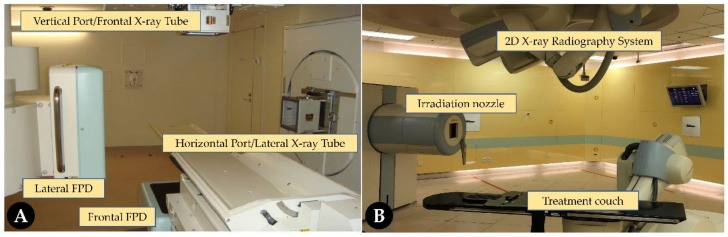
Carbon ion facilities with two-dimensional imaging systems by Shimadzu (**A**) and Siemens AG (**B**). FPD: flat panel detector; AG: Aktien Gesellschaft.

**Figure 3 cancers-11-00297-f003:**
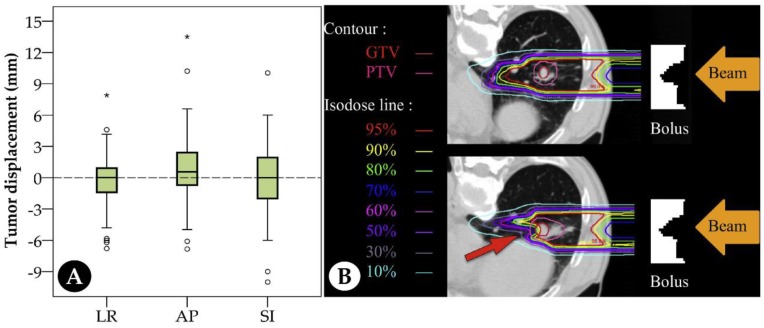
Tumor displacements of 50 patients with stage I lung cancer treated at Gunma University Heavy Ion Medical Center (**A**). * denotes the extreme value; ° denotes the abnormal value. Compared with original treatment plan (top), dose distributions significantly decreased due to internal anatomic changes in the simulation plan (bottom) (**B**). Figure 3B was adapted with permission from Irie et al. [8]. Abbreviations: LR: left–right; AP: anterior–posterior; SI: superior–inferior; GTV: gross tumor volume; PTV: planning target volume.

**Figure 4 cancers-11-00297-f004:**
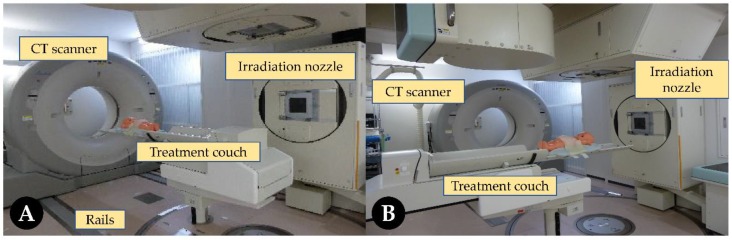
Carbon ion facility with in-room computed tomography (CT) by Hitachi and Canon Medical Systems. The treatment couch is moved for CT scanning (**A**) and then moved back for radiotherapy (**B**).

**Table 1 cancers-11-00297-t001:** The dose coverage of clinical target volume with different matching methods for particle therapy.

Authors	Year	Tumor Type	Patient Number	BM vs. TM/MM	*p*-Value
Abe S. [9]	2017	Liver	20	57.9 Gy vs. 59.8 Gy (Median D98)	0.001
Sakai M. [10]	2017	Lung	30	98.9% vs. 100% (Median V95%)	<0.001
Maeda Y. [23]	2018	Prostate	30	90.4% vs. 98.7% (Ratio of V95% > 95%)	<0.001

Abbreviations: BM: bone matching; TM: tumor matching; MM: marker matching; D98: minimum dose of covering 98% of the clinical target volume; V95%: percentage of clinical target volume receiving ≥ 95% of the dose.

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
