# Peer review of "Value of Three-Dimensional Imaging Systems for Image-Guided Carbon Ion Radiotherapy"

_cancers, 2019, doi:10.3390/cancers11030297_

Round 1

Reviewer 1 Report

I would like thank authors for their effort to answer all comments and make an appropriate revision of their manuscript.

Author Response

Dear reviewer

We are so glad that our revisions have addressed the comments. Thanks a lot for your valuable suggestions and much contribution for this manuscript.

Reviewer 2 Report

Review for Manuscript cancers-453899-peer-review-v1

General Comments: Overall, a very well-written and informative review article. I have a few specific comments that are listed below by section and line number.

More Specific Comments:

Title – None

Abstract

1)    Line 13 – Insert “a” before “smaller”

Body

1)    Line 26-28 – Reword the sentence starting with “With the”

2)    Line 28 – Insert “the” before “American”

3)    Line 62 – Change “technique” to “techniques”

4)    Line 183 – Choose a different word than “delighted”

5)    Line 183 – Change “tumor” to “tumors”

6)    Line 190-192 – Reword the sentence starting with “Gating CT”

7)    Line 216-217 – Reword the sentence starting with “However,”

8)    Line 217 – Change “system” to “systems”

9)    Line 224 – Insert “the” after “for”

10) Line 284 – Remove the word “well”

11) Line 326 – Change “profile” to “profiles”

12) Line 330 – Insert “the” before “offline”

13) Line 349 – Insert “a” before “3D”

14) Line 351 – Change “system has” to “systems have”

Author Response

Dear reviewer 

Thanks a lot for your valuable comments. We have revised the manuscript according to your suggestions.

Response to Reviewer 

Abstract

1)    Line 13 – Insert “a” before “smaller”

Response:  Revised according to comments.

Body

1)    Line 26-28 – Reword the sentence starting with “With the”

Response: We reworded the sentence: "The integration of imaging technology with the linear accelerator has led to the introduction of image-guided radiotherapy (IGRT) and the popularity of this technology has grown over recent years. "

2)    Line 28 – Insert “the” before “American”

Response:  Revised according to comments.

3)    Line 62 – Change “technique” to “techniques”

Response:  Revised according to comments.

4)    Line 183 – Choose a different word than “delighted”

Response: We replaced the word “delighted” with "successful"

5)    Line 183 – Change “tumor” to “tumors”

Response:  Revised according to comments.

6)    Line 190-192 – Reword the sentence starting with “Gating CT”

Response: We reworded the sentence: "Respiratory gated-CT, where images are collected at a certain gated window to reconstruct a CT, enables to obtain a relative stable and reproducible target position for treatment. Also, 4DCT, where CT images are reconstructed at each respiratory phase, allows a good estimation of the target motion in different respiratory phases."

7)    Line 216-217 – Reword the sentence starting with “However,”

Response: We deleted the sentence: "However, it has an unsynchronized development in particle therapy field. "

8)    Line 217 – Change “system” to “systems”

Response:  Revised according to comments.

9)    Line 224 – Insert “the” after “for”

Response:  Revised according to comments.

10) Line 284 – Remove the word “well”

Response:  Revised according to comments.

11) Line 326 – Change “profile” to “profiles”

Response:  Revised according to comments.

12) Line 330 – Insert “the” before “offline”

Response:  Revised according to comments.

13) Line 349 – Insert “a” before “3D”

Response:  Revised according to comments.

14) Line 351 – Change “system has” to “systems have”

Response:  Revised according to comments.

This manuscript is a resubmission of an earlier submission. The following is a list of the peer review reports and author responses from that submission.

Round 1

Reviewer 1 Report

The authors consider a matter of great clinical interest in radiotherapy and, particularly, in radiotherapy with ion beams. However, in my opinion, there are no original contributions in this work and much of the information provided can be found in other previous publications. Even considering that it is a review article, the authors deal with the best known aspects about the use of IGRT in radiotherapy in a superficial way, and they do not address the current research lines to develop new imaging methods in radiotherapy with proton and carbon ion beams.

In these circumstances I cannot recommend the publication of this manuscript in Cancers.

Author Response

Point 1: The authors consider a matter of great clinical interest in radiotherapy and, particularly, in radiotherapy with ion beams. However, in my opinion, there are no original contributions in this work and much of the information provided can be found in other previous publications. Even considering that it is a review article, the authors deal with the best known aspects about the use of IGRT in radiotherapy in a superficial way, and they do not address the current research lines to develop new imaging methods in radiotherapy with proton and carbon ion beams.

In these circumstances I cannot recommend the publication of this manuscript in Cancers.

Response 1: Thank you very much for your careful review and comments for this manuscript. In-room CT application is still in its infancy for carbon ion radiotherapy. It provides a big possibility for improving the accuracy of this technology. In this paper, we summarized the potential benefits from 3D imaging system and tried to push this technology forward. Sincerely hope that you can recognize the value of this article. Your recognition is greatly appreciated.

Reviewer 2 Report

Thank you for your nice review article. please find my comments here under:

What was your matrix search strategy to find relevant literature?

Line 41: You suggest : Inter and Intra-fractional correction” by using volume imaging modality. Inter fractional correction is clear but  is there any procedure / technic available for Intra-fractional online fine-tuning  dose distribution for C-ion RT such as online-ART? Would u explain it more?

Author Response

Dear reviewer,

    I am very grateful to your comments and constructive suggestions for this manuscript. According to your advice, we amended the relevant part in manuscript.

Point 1: What was your matrix search strategy to find relevant literature?

Response 1: Firstly, we selected a narrow and focused topic for this manuscript. This was helpful to search for appropriate and related literature in specific areas. Then, according to different content and purpose, articles were grouped by three types of sources as follows (Table 1). Those in primary source need comprehensive reading to collect data. Relevant literature was also easily fund in those articles. Reviews in secondary source are another good database for relevant literature. The third source contained cutting-edge research or trials. The flow chart 1. below shows the screening strategy for literature.  The key words were from primary source. Using Pubmed to search articles of interest. Next step was to select or exclude relevant literature based on specific criteria such as publish year, citations etc. Filtered articles were further classified by groups (source). The original articles as main references were analyzed carefully to get data to support the topic. To further improve the accuracy of our search, the key studies (all sources) were listed in Table 2 for evaluation [1]. Sixty-four articles (including a survey report) were finally identified by this method for the manuscript.

 Table 1The definition of three sources.

Source

Contents

     Purpose

Primary source

Current research on image-guided carbon ion therapy; related research on conventional radiotherapy.

To understand the current status of imaging technology in clinical applications and areas for improvement.

Secondary source

Reviews of IGRT including proton and photon therapy

To understand previous studies and hotspots of IGRT.

Conceptual/ theoretical

Frontier research

To know the possible direction for development.

Note: The flow chart 1. is in the attachment.

Table 2. The synthesis matrix organized by the key studies.

Author  &Date

Purpose

Method

Tumour type

Finding

Similarities

Uniqueness

Point 2: Line 41: You suggest: Inter and Intra-fractional correction” by using volume imaging modality. Inter fractional correction is clear but  is there any procedure / technic available for Intra-fractional online fine-tuning  dose distribution for C-ion RT such as online-ART? Would u explain it more?

Response 2: For intra-fractional motion management, we introduced gating-CT, 4DCT and tracking systems (Line 165-169) in this manuscript. Using field-specific target volume was also a good way to reduce the influence caused by intra-fractional variations (Line 152). Through the 4D treatment planning and gating technology, it can take intra-fractional range variations into consideration [2]. But the processing above is time consuming which means that it cannot perform online ART. However, MRI-linac enables real-time tumor tracking (Line 298 added based on your comments). And fast on-line adaptive planning based on MRI is under study in photon and proton field [3-4]. Therefore, it is feasible to achieve intra-fractional online fine-tuning dose distribution through this technology in the future. Unfortunately, there is no such research yet in carbon ion therapy. Despite MRI-guided carbon ion radiotherapy will be a very promising technology. This article can only do limited discussion on it because of few evidence.

References:

[1] A Synthesis Matrix as a Tool for Analyzing and Synthesizing Prior Research. Retrived from http://www.academiccoachingandwriting.org/dissertation-doctor/dissertation-doctor-blog iii-a-synthesis-matrix-as-a-tool-for-analyzing-and-synthesizing-prior-research.

[2] Ebner, D.K.; Tsuji, H.; Yasuda, S.; Yamamoto, N.; Mori, S.; Kamada, T. Respiration-gated fast-rescanning carbon-ion radiotherapy. Jpn. J. Clin. Oncol. 2017, 47, 80-83.

[3] Bol, G.; Hissoiny, S.; Lagendijk, J.; Raaymakers, B. Fast online Monte Carlo-based IMRT planning for the MRI linear accelerator. Phys. Med. Biol. 2012, 57, 1375-1385.

[4] Oborn, B.M.; Dowdell, S.; Metcalfe, P.E.; Crozier, S.; Mohan, R.; Keall, P.J. Future of Medical Physics: Real-time MRI guided Proton Therapy. Med. Phys. 2017, 44, e77-e90.

Reviewer 3 Report

Review for Manuscript cancers-430954-peer-review-v1

General Comments: Overall, a very well-written and informative review article. There are changes in font sizes between sections, at times. I have a few specific comments that are listed below by section and line number.

More Specific Comments:

Title – Change “system” to “systems”

Abstract

1)    Line 13 – Insert “a” before “higher” and “smaller”

2)    Line 18 – Insert “a” before “three”

Body

1)    Line 38 – Insert “a” before “three”

2)    Line 39 – Change “modality” to “modalities”

3)    Line 44 – Insert “a” before “CT”

4)    Line 48 – Insert “a” before “CT”

5)    Line 82 – Should it be “bone” or “bony”. If change to bone, change throughout the paper

6)    Line 90 – Change “limitation” to “limitations”

7)    Line 156 – Change “system” to “systems”

8)    Line 160 – Remove “very”

9)    Line 169 – Change “well” to “high”

10) Line 186 – Change “following years” to “future”

11) Line 238 – Change “plan” to “plain”

12) Line 264 – Insert “a” before “1-week”

13) Line 320 – Insert “the” before “carbon”

Author Response

Dear reviewer,

Thank you very much for your careful review and kind comments with regard to our manuscript. According to your advice, we amended the relevant part in manuscript.

Points

Title – Change “system” to “systems”

Response: Revised according to comments.

Abstract

1)    Line 13 – Insert “a” before “higher” and “smaller”

Response: Revised according to comments.

2)    Line 18 – Insert “a” before “three”

Response: Revised according to comments.

Body

1)    Line 38 – Insert “a” before “three”

Response: Revised according to comments.

2)    Line 39 – Change “modality” to “modalities”

Response: Revised according to comments.

3)    Line 44 – Insert “a” before “CT”

Response: Revised according to comments.

4)    Line 48 – Insert “a” before “CT”

Response: Revised according to comments.

5)    Line 82 – Should it be “bone” or “bony”. If change to bone, change throughout the paper

Response: All “bony ” changed to “bone ”.

6)    Line 90 – Change “limitation” to “limitations”

Response: Revised according to comments.

7)    Line 156 – Change “system” to “systems”

Response: Revised according to comments.

8)    Line 160 – Remove “very”

Response: Revised according to comments.

9)    Line 169 – Change “well” to “high”

Response: Revised according to comments.

10) Line 186 – Change “following years” to “future”

Response: Revised according to comments.

11) Line 238 – Change “plan” to “plain”

Response: Here, the CT images refer to those used in the plan.To be more precise, we changed “plan” to “planning”.

12) Line 264 – Insert “a” before “1-week”

Response: Revised according to comments.

13) Line 320 – Insert “the” before “carbon”

Response: Revised according to comments.

Round 2

Reviewer 1 Report

The authors have not made any changes to their manuscript related to my previous comments, so my opinion about this original has not changed:

"There are no original contributions in this work and much of the information provided can be found in other previous publications. Even considering that it is a review article, the authors deal with the best known aspects about the use of IGRT in radiotherapy in a superficial way, and they do not address the current research lines to develop new imaging methods in radiotherapy with proton and carbon ion beams".